# Development and Validation of a Standardized Pseudotyped Virus-Based Neutralization Assay for Assessment of Anti-Nipah Virus Neutralizing Activity in Candidate Nipah Vaccines

**DOI:** 10.3390/vaccines13070753

**Published:** 2025-07-15

**Authors:** Muntasir Alam, Md Jowel Rana, Asma Salauddin, Emma Bentley, Gathoni Kamuyu, Dipok Kumer Shill, Shafina Jahan, Mohammad Mamun Alam, Md Abu Raihan, Mohammed Ziaur Rahman, Rubhana Raqib, Ali Azizi, Mustafizur Rahman

**Affiliations:** 1International Centre for Diarrheal Disease Research, Dhaka 1212, Bangladesh; jowel.rana@icddrb.org (M.J.R.); asmageb11@gmail.com (A.S.); dipok.kumer9825@gmail.com (D.K.S.); sjahan@uams.edu (S.J.); mamun.alam@icddrb.org (M.M.A.); raihantopu007@gmail.com (M.A.R.); mzrahman@icddrb.org (M.Z.R.); rubhana@icddrb.org (R.R.); mustafizur@icddrb.org (M.R.); 2Department of Genomics and Bioinformatics, Chattogram Veterinary and Animal Sciences University, Chattogram 4202, Bangladesh; 3Medicines and Healthcare Products Regulatory Agency (MHRA), Hertfordshire EN6 3QG, UK; emma.bentley@mhra.gov.uk; 4Coalition for Epidemic Preparedness Innovations (CEPI), 0277 Oslo, Norway; gathoni.kamuyu@cepi.net (G.K.); ali.azizi@cepi.net (A.A.); 5Interdisciplinary Biomedical Sciences Program, University of Arkansas for Medical Sciences, Little Rock, AR 72205, USA

**Keywords:** Nipah virus, neutralizing antibody, pseudotyped virus, neutralization assay, validation, vaccine evaluation

## Abstract

**Background:** An effective vaccine against Nipah virus (NiV) is crucial due to its high fatality rate and recurrent outbreaks in South and Southeast Asia. Vaccine development is challenged by the lack of validated accessible neutralization assays, as virus culture requires BSL-4 facilities, restricting implementation in resource-limited settings. To address this, we standardized and validated a pseudotyped virus neutralization assay (PNA) for assessing NiV-neutralizing antibodies in BSL-2 laboratories. **Methods:** The NiV-PNA was validated following international regulatory standards, using a replication-defective recombinant Vesicular stomatitis virus (rVSV) backbone dependent pseudotyped virus. Assessments included sensitivity, specificity, dilutional linearity, relative accuracy, precision, and robustness. The assay was calibrated using the WHO International Standard for anti-NiV antibodies and characterized reference sera to ensure reliable performance. **Findings:** Preliminary evaluation of the developed NiV-PNA showed 100% sensitivity and specificity across 10 serum samples (5 positive, 5 negative), with a positive correlation to a calibrated reference assay (R^2^ = 0.8461). Dilutional linearity (R^2^ = 0.9940) and accuracy (98.18%) were confirmed across the analytical titer range of 11-1728 IU/mL. The assay also exhibited high precision, with intra-assay and intermediate precision geometric coefficients of variation of 6.66% and 15.63%, respectively. Robustness testing demonstrated minimal variation across different pseudotyped virus lots, incubation times, and cell counts. **Conclusions**: The validated NiV-PNA is a reproducible and scalable assay platform for quantifying NiV neutralizing antibodies, offering a safer alternative to virus culture. Its validation and integration into the CEPI Centralized Laboratory Network will enhance global capacity for vaccine evaluation and outbreak preparedness.

## 1. Introduction

Nipah virus was first identified during an outbreak (1998–1999) in Malaysia, where it infected 276 people and led to approximately 106 fatalities, with a case fatality rate (CFR) or 38% [1,2,3]. Due to the high CFR and lack of prophylaxis, virus culture for Nipah virus requires Biosafety Level 4 facilities. More recently, the virus has been associated with sporadic outbreaks across South and Southeast Asia, notably in Bangladesh, India, and the Philippines [3,4,5,6,7,8,9,10,11]. The CFRs in these outbreaks have typically ranged from 40 to 100% [12]. Bangladesh reported its first NiV cases in 2001; since then, NiV has caused 343 cases and 245 deaths (CFR 71%). Since 2001, India has experienced six outbreaks, while Bangladesh has reported nearly annual occurrences [13,14]. The human-to-human transmission through respiratory secretions, recurring outbreaks of the virus, high fatality rate, and the absence of vaccines or effective therapeutic treatments underscore its pandemic potential and global health risk [15].

The World Health Organization (WHO), recognizing the pandemic potential of NiV, has classified it as a priority pathogen for the development of effective vaccines and treatments [16]. There is currently no approved vaccine for NiV; however, several candidate vaccines are currently in the development stage and undergoing clinical trials [17]. Most vaccines are designed to elicit neutralizing antibodies targeting the NiV glycoprotein (NiV-G) or fusion protein (NiV-F) [18,19]. To ensure their safety, efficacy, and potential for widespread use, these vaccine candidates must undergo rigorous evaluation through clinical trials, supported by regulatory-compliant laboratory tests and immunoassays to assess immune responses to NiV.

The plaque reduction neutralization test (PRNT) and virus neutralization test (VNT) have been developed for assessing neutralizing antibodies to evaluate vaccine efficacy [20,21,22]. Both assays require high-containment (BSL-4) laboratories, limiting their accessibility, particularly in low- and middle-income countries (LMICs) and during potential pandemic situations requiring rapid assessment, like those seen during the COVID-19 pandemic [20]. To overcome this challenge, the Nipah pseudotyped virus neutralization assay (NiV-PNA) provides a safer and scalable alternative, as it can be performed in BSL-2 labs while effectively determining neutralizing antibodies against NiV [23]. The pseudotyped virus-based neutralization assay has been used as a safe alternative for performance in BSL-2 labs for a range of pathogens which require BSL-4 containment laboratories to process live viruses for neutralization assays (e.g., Rift Valley fever virus, Ebola virus, Nipah virus, Lassa virus, Nipah virus, and Marburg virus) [24,25,26,27,28,29]. In addition, a valid PNA assay can be used to explore natural immune responses against viral infections and virological changes in response to immune pressure [30,31].

In this study, a standardized and validated NiV-PNA was established. The methodology was implemented at the International Centre for Diarrhoeal Disease Research, Bangladesh (icddr,b); the technology will be transferred to selected members of the CEPI Centralized Laboratory Network (CLN), which comprises 18 partner laboratories across 13 countries. This global collaboration is expected to accelerate the development of NiV vaccines and enhance our preparedness for potential future NiV outbreaks [16].

## 2. Materials and Methods

### 2.1. Cell Lines

For pseudotyped virus production, HEK-293T/17 cells (NIBSC CFAR catalogue #5016 or ATCC CRL-11268; provided by MHRA, London, UK) were cultured in Dulbecco’s Modified Eagle Medium (DMEM; Gibco, Thermo Fisher Scientific, Waltham, MA, USA, cat. #12800-017) supplemented with 10% Fetal Bovine Serum (FBS; Gibco, cat. #16140–071), 1% L-glutamine (Sigma-Aldrich, Saint Louis, MO, USA, cat. #G8540), and 2% HEPES (Sigma-Aldrich, cat. #H3375).

For NiV-PsV titration and neutralization assay, Vero cells (African green monkey kidney cells, ATCC CCL-81) were used. The Vero cells were cultured in DMEM supplemented with 10% FBS, 0.8% Fungizone (Amphotericin B; Sigma-Aldrich, cat. #A9528), 1% penicillin/streptomycin (Gibco, cat. #15140–122, USA), 1% L-glutamine (Sigma-Aldrich cat. #G8540-100G), 1% sodium pyruvate (Sigma-Aldrich cat. #P2256-100G), and 2% HEPES. Cells were incubated at 37 °C with 5% CO_2_ and passaged every 3 to 4 days in 100 mm TC-treated cell culture dishes (Corning, New York, NY, USA, cat. #430167). For assay purposes, the same medium was used with 5% FBS.

### 2.2. Sample Panel and Reference Standards

This study utilized the WHO International Standard (WHO IS) for anti-Nipah virus (anti-NiV) antibodies in neutralization assays (NV-1; NIBSC product code: 22/130_NT), alongside five other reference sera containing anti-NiV antibodies (NV-2, NV-3, NV-4, NV-6, and NV-10); in addition, we prepared NNV-1, NNV-2, NNV-3, and NNV-4 by diluting NV-1 (Appendix A) [32]. NV-1 served as the high positive assay control, and NV-3 served as the low positive assay control. Five negative sera (NC-1, NC-2, NC-3, NC-4, NC-5) collected at icddr,b were confirmed to be free of NiV exposure using reverse-transcriptase polymerase chain reaction (rt-PCR) and ELISA (NiV-G protein) following CDC (Centers for Disease Control and Prevention, Atlanta, GA, USA) protocols [33,34] (Appendix A).

### 2.3. Plasmid Preparation

The plasmids for the NiV-PsV production, containing codon-optimized NiV-G and NIV-F genes from the NiV Bangladesh isolate (NCBI accession number: JN808864.1) were generously provided by Dr. Edward Wright (University of Sussex, Brighton, UK) and Professor Teresa Lambe (Oxford Vaccine Group, Oxford, UK), respectively. These plasmids were transformed into competent *E. coli* JM109 cells (Promega, Madison, WI, USA, cat no. #L2001) and then plasmid DNA was extracted using a Qiagen midiprep kit (Qiagen, Hilden, Germany, cat. #12145) [35,36].

### 2.4. Pseudotyped Virus Production

The pseudotyped virus was prepared using an rVSV backbone, in which the VSV glycoprotein (G) gene was replaced with a firefly luciferase (Luc) reporter gene, based on a method originally described by Dr. Michael A. Whitt in 2010 [37]. In brief, for the NiV-PsV production, HEK-293T/17 cells were cultured in 100 mm TC-treated dishes (4 × 10^6^ cells/dish) (Corning, cat. #430167) overnight at 37 °C with 5% CO_2_. The next day, once cells reached 50–60% confluency, they were co-transfected with pCAGGS NiV-G and pCAGGS NiV-F expression plasmids using the transfecting agent Polyethylenimine (PEI) (Sigma-Aldrich, cat. #408727). Approximately (20 ± 2) hours post-transfection, the cells were infected with rVSV*ΔG virus (Kerafast, Boston, MA, USA, cat. #EH1020-PM) at a multiplicity of infection (MOI) of 2 (Appendix A). Following (18 ± 2) hours post-infection, NiV-PsV particles were harvested (Figure 1).

Then, the harvested NiV-PsV were filtered through a 0.45 µm syringe filter (Globe Scientific, Mahwah, NJ, USA, cat. #SF-CA-4530-S). To further concentrate and remove cellular debris, the filtrate was centrifuged on a 20% (g/v) sucrose cushion (Fisher Scientific, Waltham, MA, USA, CAS 57-50-1) in a round-bottom, screw-cap 10 mL centrifuge tube (Tarsons, Kolkata, West Bengal, India, cat. #TARST541020) at 20,000× *g* for 6 h at 4 °C. The pelleted NiV-PsV was resuspended in 3 mL PBS and stored at −80 °C in aliquots until further use [23,37].

### 2.5. Pseudotyped Virus Titration

The 50% tissue culture infectious dose (TCID_50_) was determined to quantitate the pseudovirus particle. Vero cells (2 × 10^4^ cells/well) were seeded in a TC-treated 96-well flat-bottomed microplate (Corning, cat. #3917) and incubated at 37 °C with 5% CO_2_ overnight (80–90% confluent). The next day, the serial dilution of NiV-PsV virus was performed in a 96-well V-bottom microplate (Corning, cat. #3894) according to plate layout (Appendix A). After 1 h incubation, the serially diluted virus was transferred onto the microplate pre-seeded with Vero cells and incubated overnight (20 ± 2 h) at 37 °C with 5% CO_2_. Following incubation, the supernatant was carefully aspirated using an aspirator (SCILOGEX SCIVac Vacuum Aspirator, Rocky Hill, CT, USA, cat. #761000019999). Subsequently, 50 μL of a 1:1 mixture of ONE-Glo™ EX Luciferase assay reagent (Promega, cat. #E8130, USA), and assay medium was added to each well. Immediately, plates were incubated in a microplate shaker at 600 rpm for 3 min at room temperature, and luminescence was measured in relative luminescence units (RLUs) using a multimode plate reader (PerkinElmer Victor Nivo^TM^, Waltham, MA, USA, serial number: HH35L1122076). The TCID_50_ of NiV-PsV was determined using the Reed–Muench method [38].

### 2.6. Pseudotyped Virus-Based Neutralization Assay

The PNA was developed to assess NiV-specific neutralizing antibodies in human serum. Heat-inactivated sera were serially diluted 2-fold in a V-bottom microplate (plate map in Appendix A) and NiV-PsV (100 TCID_50_/well) was added to each well, except for cell controls. All the serum samples were tested in duplicate in each plate. Plates were incubated at 37 °C with 5% CO_2_ for 1 h and the serum–PsV complexes were transferred onto the previously seeded microplates with Vero cells (2 × 10^4^ cells/well). Following incubation at 37 °C with 5% CO_2_ for approximately 20 h, luciferase substrate was added, and plates were read on a luminescence microplate reader. The luminescence intensity inversely correlated with the neutralizing antibody present in the serum. Neutralization titers were determined using a 4-parameter logistic regression curve, which represents the dilution required to reduce the RLU signal by 50% relative to that of the virus control in the absence of serum (Figure 2).

### 2.7. NiV-PNA Validation

Neutralization assay validation was performed in accordance with the “United States Pharmacopeia (USP) Convention General Chapter on Biological Assay Validation (<1033>)” and the “ICH Harmonized Tripartite Guideline Validation of Analytical Procedures: Text and Methodology Q2(R1)”, utilizing validated instruments and software [39,40,41]. The potency of the samples was measured in International Units per milliliter (IU/mL), relative to the WHO International Standard (NV-1) for each assay run (Appendix A).

Sensitivity and specificity assessments were conducted using a panel of 5 positive and 5 negative human serum samples for anti-NiV antibodies (Appendix A). And these 5 positive samples considered to be true positive based on the results of the collaborative study [32]. Inter-laboratory specificity was evaluated by comparing measured results of these 5 positive samples between the two laboratories (icddr,b and MHRA) performing NiV-PNA.

Dilutional linearity was evaluated using a two-fold dilution series (1:1 (neat) to 1:32) of three anti-NiV antibody-positive serum samples (Appendix A). Geometric mean titers (GMTs) were calculated for each dilution. A linear regression model was applied to compare serum dilution to GMT, with slope and R^2^ used to assess linearity. Relative accuracy was determined by comparing the observed titers to expected titers (Appendix A).

To assess the precision, a panel of 10 human serum samples (8 positive and 2 negative) were assessed in duplicate by two analysts over three different days (in parallel). The precision parameters were assessed using the percentage geometric coefficient of variation (%GCV) [32,42] (Appendix A).

The lower limit of quantification (LLOQ) of this assay was established by generating a calibration curve using a positive human serum sample for anti-NiV neutralizing antibodies. A two-fold serial dilution was performed, spanning from 1:1 (neat) to 1:4096, with each dilution tested in duplicate. The detection limit for neutralizing anti-NiV antibodies in human serum was calculated based on the standard deviation of the response and the slope of the calibration curve [43] (Appendix A).

The robustness of the assay was evaluated on three anti-NiV antibody-positive serum samples by modifying several parameters: (i) the number of Vero cells in the assay plate (15,000, 18,000, 20,000, 22,000, and 25,000 cells per well), (ii) the use of two different NiV-PsV lots (Lot-1 and Lot-2), and (iii) variations in the incubation period on day 2 of the NiV-PNA (18, 20, 22, and 24 h). Sample variability was evaluated using the % GCV (Appendix A).

### 2.8. Statistical Analysis of Data

Neutralizing antibody titers in IU/mL unit was determined from the RLUs data using a 4-parameter logistic regression curve, calculated with SoftMax Pro GxP Compliance Software Suite (version 7.1.1). Experimental data were compiled and analyzed by Microsoft Excel 2019. After that, the experimental outcomes and analyzed data were visualized by GraphPad Prism (version 9). MedCalc program (version 22.016) was used to determine the sensitivity and specificity of the assays.

## 3. Results

### 3.1. NiV Pseudotyped Virus Particle Production

The harvested NiV-PsV had a titer of >10^7^ TCID_50_/mL. NiV-PsV titer variability was monitored over a 12-month period (August 2023–July 2024). The overall titer (TCID_50_/mL) remained within ±3 standard deviation (SD) throughout the timeline (Figure 3).

### 3.2. Validation of NiV-PNA

To evaluate the performance of the NiV-PNA using Vero cells, key validation parameters were assessed based on the established acceptance criteria outlined in the previously published studies and relevant guidelines [39,40,44]. All criteria were met, confirming the reliability and accuracy of the assay under the tested conditions. A summary of the validation outcomes is provided in Table 1.

#### 3.2.1. Specificity and Sensitivity

The NiV-PNA showed preliminary 100% sensitivity and specificity in inter-laboratory testing. All positive samples (n = 5) were correctly identified as positive, and all negative samples (n = 5) as negative, with a 95% Confidence Interval (CI) of (47.82–100)% (Figure 4a). The inter-laboratory comparison of five positive samples between icddr,b and MHRA yielded an overall geometric mean ratio (GMR) of 0.93, with individual GMRs ranging from 0.64 to 1.13 (Figure 4b, Appendix A). The coefficient of determination (R^2^) of 0.8461 was observed between both calibrated assays (Figure 4c).

#### 3.2.2. Dilutional Linearity

Dilutional linearity was assessed by evaluating the linear regression slope between measured and expected values, with an acceptable range defined as 0.80–1.25 [44]. The assay demonstrated a linear relationship between sample potency and dilution factor within the analytical range of 11–1728 IU/mL (Appendix A). The R^2^ values were 0.9969, 0.9943, and 0.9959 for NV-2, NV-4, and NV-10, respectively, with an overall regression slope of 1.04, confirming strong linearity (Figure 5).

#### 3.2.3. Relative Accuracy

An assay is considered to have acceptable relative accuracy if the geometric mean (GM) recovery falls within the range of 70–130% [44]. Across the tested anti-NiV antibody titer (11–1728 IU/mL) for samples NV-02, NV-04, and NV-10, the GM recovery was 98.18%. All dilutions met the accuracy criteria, with relative accuracy ranging from 76.90% to 128.47% (Appendix A).

#### 3.2.4. Precision

Precision was evaluated by measuring the percent geometric coefficient of variation (%GCV). According to US Food and Drug Administration (US-FDA) method validation guidelines, a precision of ≤20% is acceptable for most concentrations, with ≤25% allowed at the LLOQ [45]. Considering the inherent variability of cell-based pseudotyped virus neutralization assays, a secondary acceptance criterion of ≤30% was pre-specified based on established regulatory guidance and supporting literature [41,44,46]. Intra-assay precision (repeatability) yielded an average GCV of 6.66% for the precision panel, while the intermediate precision (total variability), showed a GCV of 15.63% (Table 1). Overall, both the intra-assay precision and intermediate precision are within the threshold (≤30%), confirming that the method meets acceptance criteria.

#### 3.2.5. Lower Limit of Quantification

The minimum detectable concentration was determined using a calibration curve generated by serially diluting sample NV-4 (neat 1:1 to 1:4096), which is selected due to its high titer among the five positive samples. The calibration curve exhibited a slope (measured vs. expected titer) of 1.0007 and R^2^ value of 0.9978, indicating strong linearity (Figure 6). The LLOQ was calculated to be 2.78 IU/mL, based on the standard deviation of the response and the slope.

#### 3.2.6. Robustness

Minimal variation was observed when different pseudotyped virus lots were used, with a GCV of 4.03%, which is low considering the inherent variability of cell-based assays [47]. Variations in incubation time after adding cells to the pseudotyped virus–serum mixture resulted in a GCV of 20.90%, while variations in cell number per well led to a GCV of 27.57%. However, when the cell count was limited to 18,000, 20,000, and 22,000 per well, the GCV decreased to 10.65% (Figure 7), indicating improved consistency under controlled conditions.

## 4. Discussion

We successfully standardized and validated a PNA to assess the neutralizing potency of serum antibodies against NiV, ensuring compliance with international guidelines. This rigorous approach ensures the assay’s reliability and reproducibility for use in serological studies and vaccine evaluation.

To develop the assay, an rVSV-ΔG backbone-based Nipah pseudovirus was utilized. This system was chosen over the lentiviral system due to its ability to generate high-titer PsV particles and high-throughput, ensuring consistency and reliability [23,48,49]. Previous studies by Negrete et al. and Kaku et al. have utilized this rVSV-ΔG platform to investigate the NiV cell entry and serological properties, by incorporating the NiV-G and NiV-F proteins into the pseudotyped virus [50,51]. Negrete et al. primarily used NiV pseudotyped viruses to investigate viral receptors. Kaku et al. developed a VSV-based pseudotyped neutralization assay for NiV and validated using non-human serum samples. In contrast, we first report validation of VSV-based pseudotyped neutralization assay using human serum samples and the WHO International Standard (WHO IS) for anti-NiV antibodies. This distinction enhances the clinical relevance of our study, particularly for human vaccine evaluation and sero-surveillance. Additionally, Cai et al. validated a pseudotyped virus neutralization assay to evaluate neutralizing antibodies against SARS-CoV-2 and its Omicron subvariants [52]. Nie et al. developed Rabies virus neutralization assays, demonstrating high correlation with traditional methods like Rapid Fluorescent Focus Inhibition Test (RFFIT), while offering greater sensitivity and faster results [53]. Besides, pseudotyped viruses have been used for CHIKV DNA vaccine evaluation and LASV neutralization assays, aiding vaccine development and comparison studies [46,54].

The WHO Expert Committee on Biological Standardization established an WHO IS for anti-NiV antibody based on collaborative study involving 18 laboratories [32]. This serves as a key reference for harmonizing serological assays and ensuring the comparability of results across studies. By calibrating the NiV-PNA against the WHO IS, neutralizing antibody titers can be quantified in International Units (IUs), enabling direct comparisons across different studies and platforms in clinical vaccine trials.

The assay demonstrated a positive correlation between neutralizing antibody titers measured in serum samples by the collaborative study and the icddr,b laboratories, confirming its accuracy. Inter-laboratory comparisons between icddr,b and MHRA further demonstrated that the validated assay has good concordance with other calibrated NiV-PNA.

The sensitivity and specificity results, based on a limited panel of well-characterized samples, showed accurate identification of true positives, as reported in the collaborative study, supporting preliminary evidence of the assay’s reliability in detecting anti-NiV neutralizing antibodies. The dilutional linearity assessment confirmed that the assay generates consistent results across a range of serum dilutions, supporting its quantitative reliability. Additionally, relative accuracy measurements revealed a correlation between observed and expected neutralizing antibody titers, ensuring precise titer determination. Precision analysis, evaluated through intra-assay and total variability assessments, indicated minimal variation, further supporting the assay’s reliability across different experimental conditions.

Robustness assessments showed that the assay generally maintained performance within acceptable limits under variations in standard laboratory conditions. While changes in pseudotyped virus batches and processing time resulted in relatively low variability, alterations in cell seeding density led to increased %GCV, indicating that certain parameters—particularly cell number—can influence assay readout. These findings suggest that, although the assay tolerates some procedural variation, careful control of specific conditions, such as cell density, is important to ensure consistency across laboratory settings.

Despite the successful standardization and validation of the NiV-PNA, certain limitations should be acknowledged, principally due to limited number and amount of specimen availability from NiV infection survivors. Specifically, we were unable to assess the impact of multiple freeze–thaw cycles on serum stability, nor the potential interference from lipemic or hemolytic samples. Only serum samples were used, and comparison between plasma and serum matrices were not performed. The upper limit of quantification (ULOQ) remains undermined, also due to the limited availability of serum samples. Future studies using a larger and more diverse panel, including vaccinee serum and plasma, are critically needed to address current limitations and to generate more robust data. Considering the limitation of human sera availability, we recommend standardizing other experimental animal sera (e.g., swine) for validation purposes. Additionally, comparing the NiV-PNA with live virus neutralization assay using wild-type NiV, optimizing different antifungals with less membrane disruption in growth medium will help to optimize the method. Such a comparative analysis would provide valuable insights into its correlation with the gold-standard method. Future studies should prioritize this comparative analysis to determine whether both assays yield com-parable neutralization profiles. Although the emergence of replication-competent virus is unlikely, we did not assess this experimentally; conducting such a risk assessment would further strengthen the safety evaluation of the assay.

## 5. Conclusions

Despite current limitations, this study addresses a critical challenge in NiV research in Bangladesh. The development and validation of NiV-PNA in a country with recurrent NiV outbreaks, which is considered a potential epicenter for future spillover events, holds significant public health relevance. The assay’s ability to determine neutralizing antibodies without requiring live virus enhances its utility in resource-limited and non-BSL-4 settings, making it a valuable tool for global health preparedness.

Looking forward, we plan to optimize the NiV-PNA for additional NiV variants by incorporating alternative glycoproteins. Given the antigenic diversity among NiV strains, such optimization will support comprehensive evaluation of immunogenic variability, cross-neutralization potential, and vaccine efficacy. This expansion will enhance our understanding of immunity across different NiV lineages and guide public health strategies.

The integration of this standardized and validated NiV-PNA into the CEPI-CLN represents a significant advancement in enhancing global health preparedness, facilitating robust vaccine evaluation and outbreak response capabilities. Future studies should explore its application in large-scale epidemiological studies and clinical trials to further strengthen its utility in NiV research and public health preparedness.

## Figures and Tables

**Figure 1 vaccines-13-00753-f001:**
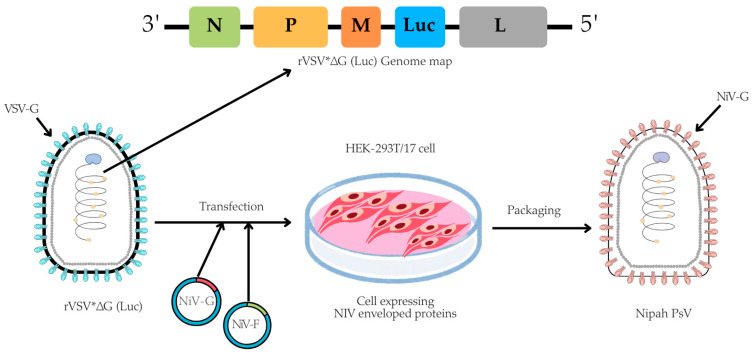
A schematic illustration of NiV-PsV production. The genome map of recombinant Vesicular stomatitis virus (rVSV*ΔG), which lacks the VSV envelope G protein and contains an additional firefly luciferase gene (Luc) as a marker, is transfected into cells expressing the Nipah envelope proteins (glycoprotein, NiV-G, and fusion, NiV-F). Created using Canva, including elements from the NIH BioArt Vector Library (https://www.bioart.nih.gov).

**Figure 2 vaccines-13-00753-f002:**
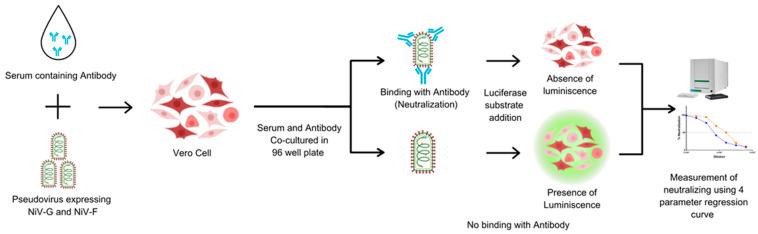
The principle of the pseudotyped virus neutralization assay (PNA). Vero cells are incubated with NiV-PsV in the presence of serum. In the absence of neutralizing antibodies, the pseudovirus infects the cells. The addition of substrate results in luminescence. The luminescence signal is detected, and the presence of neutralizing antibodies is quantified using a four-parameter regression curve. Created using Canva, including elements from the NIH BioArt Vector Library (https://www.bioart.nih.gov).

**Figure 3 vaccines-13-00753-f003:**
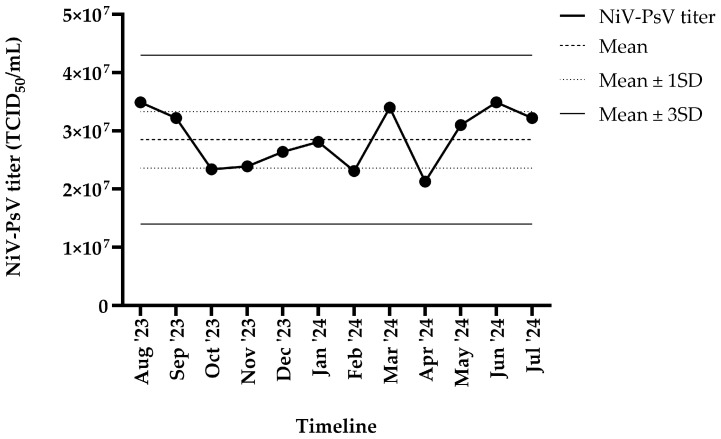
NiV-PsV titer variability over 12 months period. The *X*-axis represents the timeline, and the *Y*-axis indicates the NiV-PsV titer. The horizontal dashed line represents the mean titer, the dotted line indicates the mean ± 1 SD, and the solid line represents the mean ± 3 SD.

**Figure 4 vaccines-13-00753-f004:**
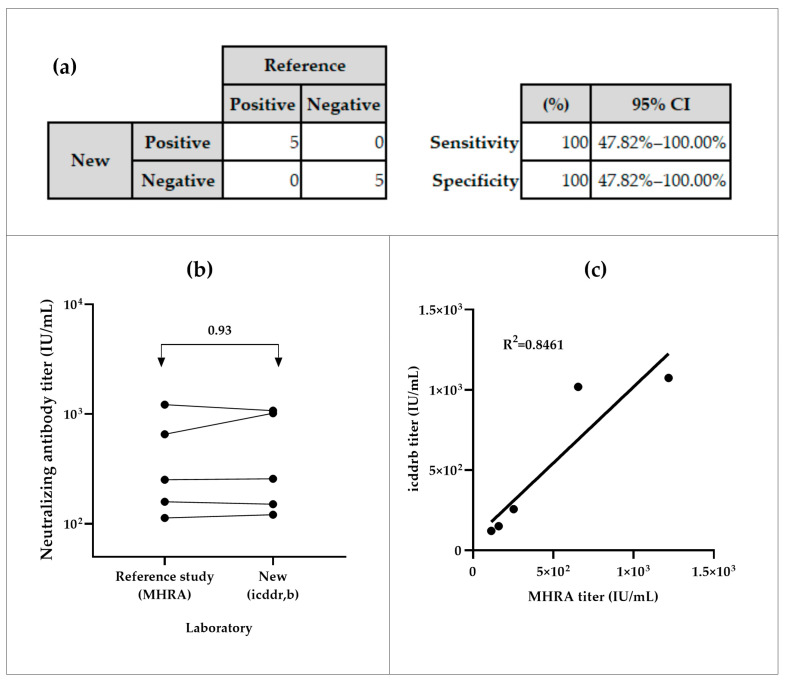
Specificity and sensitivity analyses of serum samples from icddr,b and MHRA. (**a**) Sensitivity of the NiV-PNA assay, based on MedCalc metrics. (**b**) Geometric mean ratios (GMR) of neutralizing antibody titer (IU/mL) from inter-laboratory comparison between MHRA and icddr,b. (**c**) Correlation analysis of measured neutralizing antibody titers from the specificity panel.

**Figure 5 vaccines-13-00753-f005:**
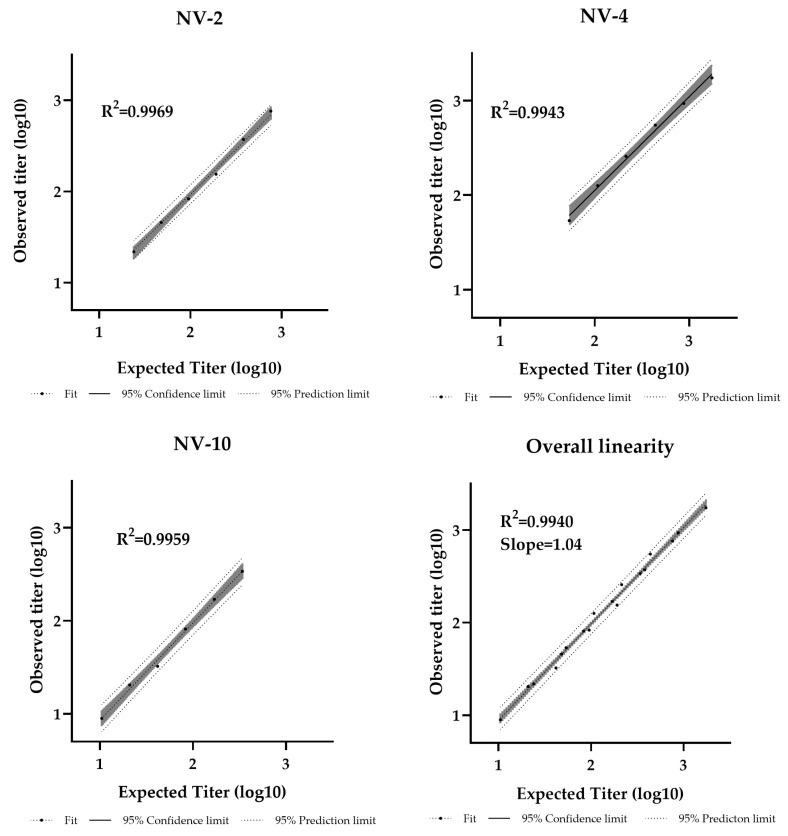
Pseudotyped virus neutralization assay dilutional linearity of high and low titer samples. High-titer samples (NV-2, NV-4) and a low-titer sample (NV-10) were tested neat and at 1/2, 1/4, 1/8, 1/16, and 1/32 dilutions. Overall regression slope is 1.04 and R^2^ = 0.9940.

**Figure 6 vaccines-13-00753-f006:**
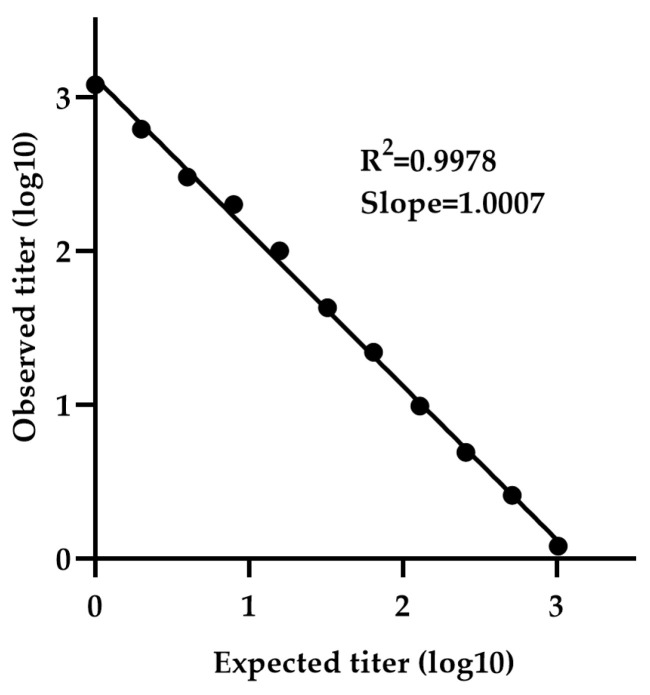
Determination of LLOQ by NiV-PNA using serial dilutions of NV-4. The minimum detectable concentration was determined using a calibration curve generated by serially diluting NV-4 (1:1 to 1:4096). The calibration curve exhibited a slope (measured vs. expected titer) of 1.0007 with an R^2^ value of 0.9978.

**Figure 7 vaccines-13-00753-f007:**
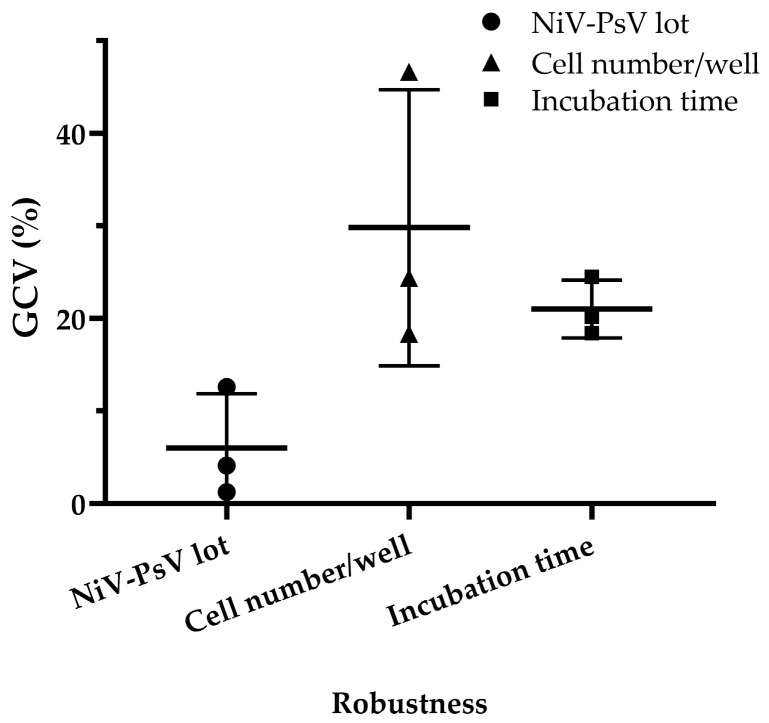
NiV-PNA robustness: Effects of NiV-PsV lot variability, cell number per well, and incubation time on %GCV for NNV-4, NV-2, and NV-4 sera. Robustness was assessed using three anti-NiV antibody-positive serum samples by varying the following: (i) NiV-PsV lot (Lot-1 and Lot-2, GCV = 4.03%), (ii) the number of Vero cells per well (15,000–25,000, with highest variability at 25,000, GCV = 27.57%; reduced to 10.65% when limited to 18,000–22,000), and (iii) incubation period on day 2 of NiV-PNA (18–24 h, GCV = 20.90%).

**Table 1 vaccines-13-00753-t001:** Acceptance criteria for each validation parameters for validating Nipah pseudotyped virus neutralization assay using Vero cell.

Parameter	Acceptance Criteria	Validation Outcome	Passed/Failed
Sensitivity	≥80%	100%	Passed
Specificity	≥80%	100%	Passed
Dilutional linearity	Linear regression slope (GMT ^1^, observed vs. expected) 0.80–1.25 and R^2^ ≥ 0.95	Slope = 1.04; R^2^ = 0.9933	Passed
Relative accuracy	percent recovery: (70–130)%	98.18%	Passed
Intra assay precision (Repeatability)	GCV ^2^ ≤ 30%	6.66%	Passed
Intermediate precision (Total variability)	GCV ≤ 30%	15.63%	Passed

^1^ GMT, geometric mean titer. ^2^ GCV, geometric coefficient of variation.

## Data Availability

All data presented in this study are available in the article and Appendix A.

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
