# Peer review of "Development and Validation of a Standardized Pseudotyped Virus-Based Neutralization Assay for Assessment of Anti-Nipah Virus Neutralizing Activity in Candidate Nipah Vaccines"

_vaccines, 2025, doi:10.3390/vaccines13070753_

Round 1

Reviewer 1 Report

Comments and Suggestions for Authors

The authors describe the development and validation of a high throughout neutralizing antibody assay for Nipah virus that utilizes pseudovirus technology to enable assay performance without the need for BSL-4 containment facilities. The assay was partially optimized in terms of virus dose, cell numbers and incubation time. Validation parameters included sensitivity, specificity, linearity, precision and accuracy; assay results met pre-specified pass criteria for all parameters. The study was limited by a relatively small number and volume of available serum samples containing anti-Nipah virus antibodies. Overall the manuscript is well-written, and the assay is described in adequate detail to be reproduced in other laboratories.

Major point:

The assay was developed for the main purpose of enabling neutralization assays in BSL-2 containment facilities rather than in BSL-4 facilities that are required for work with the live virus. This assumes the pseudovirus is not replication competent and therefore safe for use in BSL-2 labs. While one can argue based on the genetics of the platform that no replication competent virus should be present, nature has a way of being unpredictable. Given the highly pathogenic nature of Nipah virus, it seems premature to recommend the described pseudovirus assay for BSL-2 laboratories without first testing for the presence of replication-competent virus in pseudovirus preparations. This will of course require collaboration with a BSL-4 facility.

Other points to address:

  1. I noticed the growth medium used in the assay is supplemented with the antifungal drug amphotericin B. This drug is known to inhibit enveloped viruses and in fact is quite potent against HIV. The authors may want to test whether it inhibits their Nipah pseudoviruses and consider omitting it from the growth medium if necessary.

  1. Line 99: Should NV-3 be NV-4?

  1. Line 128-132: What is the purpose centrifugation on a sucrose cushion as a final step in pseudovirus preparation? Is this step needed?

  1. Lines 142-144: How is the Luc gene regulated? Is it constitutive expression?

  1. Section 3.1.5. LLOQ: Limits of quantitation are usually defined as titers that can be measured with acceptable linearity and precision. Can you say anything about the precision of the assay at the LLOQ of 2.78 IU/ml?

  1. Please define icddr,b

Reviewer 2 Report

Comments and Suggestions for Authors

Please see my detailed comments attached.

Reviewer assessment summary:   This manuscript presents a scientifically rigorous study describing the development and validation of a PNA for NiV. The methodology is sound and appropriately described in detail. The manuscript is well structured and exceptionally well written, both of which enhances  clarity and reader engagement.  Beyond technical and structural strengths, the study also has clear relevance to epidemic preparedness and international vaccine evaluation efforts. The systematic validation approach and the proposed connection to the CEPI Centralized Laboratory Network underscore the potential real-world impact of this work.  That said, there are a few areas where the manuscript could be improved. Most notably, the very limited size of the validation panel does not support strong performance claims for sensitivity and specificity. This constraint should be more clearly framed as a significant limitation. Some additional engagement with previous literature would also help to differentiate this study from directly relevant published work, emphasizing its strengths and novelty. Overall, this manuscript clearly falls within the top tier of assay development studies that I have reviewed, and I commend the authors for their careful preparation and clear presentation. I hope my comments attached will help the authors further elevate the quality of their manuscript, which is already well positioned to make a valuable contribution to the field.

Round 2

Reviewer 1 Report

Comments and Suggestions for Authors

I am concerned the authors are not taking seriously the need to test for replication-competent virus before recommending this assay for BSL-2 level work. Their pseudovirions contain two components of Nipah virus that are determinants of virulence. The authors should remove their recommendation for BSL-2 unless they perform a robust tests for RCV. 

Author Response

Comments and Suggestions for Authors

I am concerned the authors are not taking seriously the need to test for replication-competent virus before recommending this assay for BSL-2 level work. Their pseudovirions contain two components of Nipah virus that are determinants of virulence. The authors should remove their recommendation for BSL-2 unless they perform a robust tests for RCV.

Response: We understand the reviewer’s concern given the pathogenicity of Nipah virus. However, a substantial body of literature supports using pseudoviruses at BSL-2 to study pathogens requiring BSL-4 containment. The expression of Nipah virus F or G proteins alone cannot biologically cause Nipah virus infection, as these systems lack all other genes necessary for Nipah virus generation. Furthermore, the VSV backbone manufacturer (Kerafast) confirms this system is approved for BSL-2 handling. While we propose assessing the theoretical reversion risk of pseudotype viruses (as noted in the revised manuscript), the extensive existing literature suggests this concern is biologically unlikely to happen. For the reviewer’s and editor’s reference, we have included a list of more supporting publications which explicitly mention that PNA can be performed in BSL-2 laboratories for different purpose. Among these we have cited seven more articles including authors own work on pseudoviruses in the revised manuscript. 

  • Rift Valley fever virus: https://www.tandfonline.com/doi/10.1080/21645515.2019.1627820#abstract
  • Ebola: https://www.sciencedirect.com/science/article/pii/S0264410X17303250
  • SARS-CoV-2, EBOV, LASV, CHIKV and VSV: https://www.mdpi.com/1999-4915/12/12/1457
  • Nipah: https://www.sciencedirect.com/science/article/pii/S004268222300171X
  • Marburg virus: https://www.frontiersin.org/journals/microbiology/articles/10.3389/fmicb.2022.927122/full
  • CHIKV: https://pmc.ncbi.nlm.nih.gov/articles/PMC3647917/
  • Review: https://www.tandfonline.com/doi/epdf/10.1080/14760584.2023.2299380?needAccess=true
  • Kerafast statement: https://www.kerafast.com/cat/122/delta-g-vsv-pseudotyping-system

Revised manuscript (Line 72-77): Pseudotyped virus-based neutralization assay has been used as a safe alternative to perform in BSL-2 labs for a range of pathogens which require BSL-4 containment laboratory to process live viruses for neutralization assay (e.g., Rift Valley fever virus, Ebola virus, Nipah virus, Lassa virus, Nipah virus, Marburg virus) [24–29]. In addition, a valid PNA assay can be used to explore natural immune response against viral infections and virological changes in response to immune pressure [30,31].

Round 3

Reviewer 1 Report

Comments and Suggestions for Authors

With the recent changes, the manuscript is now suitable for publication.